# Disgusting or Innovative-Consumer Willingness to Pay for Insect Based Burger Patties in Germany

**Lukas Kornher [1],\* [ID], Martin Schellhorn [2] and Saskia Vetter [2]**

1    Center for Development Research (ZEF), University of Bonn, 53113 Bonn, Germany
2    Department of Food Economics and Consumption Studies, Kiel University, 24118 Kiel, Germany;
     mschell@food-econ.uni-kiel.de (M.S.); saskiavetter@aol.com (S.V.)
\*    Correspondence: lkornher@uni-bonn.de

**Abstract:** Insects represent an excellent source of food due to their density in unsaturated fatty acids, vitamins, and minerals, while their production is associated with lower emissions of greenhouse gases and resource use as compared to other conventional protein sources. In most Western countries, the human consumption of insects is very low and often perceived as culturally inappropriate. In this study, we analyzed the preferences of German consumers for insect-based products to intensify the knowledge about specific consumer segments that are willing to adopt insects into their diet. For this purpose, an online based choice experiment was conducted in 2016, in which respondents chose between an ordinary burger and a burger with a beef burger patty fortified with insect flour. We detect three homogeneous consumer segments in our sample. The largest group of respondents is willing to consume insect-fortified burgers with only a small price discount, while the other respondents had a prohibitively low willingness-to-pay. The readiness of consumers to adopt insects into their diet is strongly related to attitudinal variables, such as preferences for an environmental friendly production method and health aspects. On the other hand, disgust and the aversion towards insects seem to be the main reasons to abstain from eating insects.

**Keywords:** edible insects; latent class model; willingness-to-pay; Germany

---

## 1. Introduction

The large scale cultivation of animals to satisfy global demand for meat products has severe consequences on greenhouse gas emissions and resource use. The growing demand for animal proteins through rising incomes in middle income countries, like India, China, and Brazil, will exacerbate the situation in the coming years. In addition to that, frequent and high consumption of meat products is considered to increase the risk of cardiovascular diseases and obesity [1].

These accompanying symptoms of meat consumption have increased the importance of the discussion and research on insects in human nutrition [2]. Insect farming, also known as micro livestock, has several advantages as compared to other cultivated food animals. From the perspective of nutrition physiology, insects represent an excellent source of food due to their density in unsaturated fatty acids, vitamins, and minerals [3], while the level of protein is similar to beef and pork meat [4]. In addition, there are ecological benefits, which include lower emissions of greenhouse gases, lower resource use of land and water, a higher feed conversion efficiency, and the great potential to transform low value by-products into high quality food products in comparison to conventional meat sources, in particular red meat [5,6].

During early development stages of mankind, before hunting and farming of animals was undertaken, insects were an important part of the human diet. In the absence of vertebrates in warmer areas, the human consumption of insects has been carried to modern times [7]. Today, over 2000 insect

species are known to be edible, most commonly consumed are caterpillars, crickets, bugs, grasshoppers, and ants [8]. According to the FAO [9], two billion people in more than 100 countries regularly consume insects, while the rates are highest in Africa, Asia, and Latin America.

However, acceptance rates of insects as foodstuff by consumers in Western countries are generally low [10–12]. For this reason, for a long time, it was hardly possible to purchase insect-based products in normal grocery stores on a regular basis, and those products were only offered in delicatessen restaurants, specialized shops, and over the internet [13]. This has changed only a few years ago. In August 2017, the Swiss food retailer, Coop, offered insect-based burgers in their shops and online. After positive costumer reactions, Coop started to also sell cricket based energy bars. Since the beginning of 2018, the new Novel Food Regulation by the European Union explicitly lists insects in the Novel Food Catalogue and changes the authorization process for novel foodstuffs. Applications for new food products can be sent to the European Food Safety Authority and foodstuffs need to be tested regarding their health safety before approval, which makes the process more transparent [14]. In April 2018, the German producer, Bugfoundation, started to market its insect fortified burger, made of buffalo worms and organic soy, through the retailer, REWE, in Germany.

Given the limited experience with and data for insect based food products, understanding consumer behavior is crucial to inform food processing companies about the potential target group of their products. Based on this knowledge gap, this study, which profiles consumers who are ready to adopt and identifies the factors explaining consumers' reluctance to adopt insects in their diet, is a first step into this direction [12]. This includes psychological and sensory factors, such as disgust towards eating insects and food neophobia, which is the aversion towards new and unknown food products [15,16], but also the cultural environment in Western countries, which sees insect consumption as inappropriate or even taboo [17]. Further, we adopt a conjoint based choice experiment to elicit the willingness-to-pay of the respondents for insect based burger patties taking into account the possible heterogeneity of the respondents.

According to Ruby et al. [18], the potential adopter in the United States has a low level of disgust and higher 'sensation seeking' traits. Other studies argue that the familiarity with eating insects and a high preference for convenience products contribute a greater willingness to adopt [12,19,20]. In a study among German consumers, Hartmann et al. [19] find that consumers with low levels of food neophobia, high scores for social acceptance of insects, and positive taste expectations reported a significantly higher willingness to consume insects. In line with this, we can expect that psychological factors are a main driver of the dislike towards insects, which will cause a general aversion towards purchasing insect-based products irrespective of their attributes.

Environmental sustainability is reported to be the most beneficial attribute of insect-based products to consumers [12,18]. This supports findings from studies on the preferences for the environmentally friendliness of production methods, such as Grunert et al. [21], who show that consumers in Germany and Poland are concerned about the environmental impact of meat cultivation and are willing to pay for reduced carbon emissions. The beneficial environmental characteristics of insects clearly relate to adverse characteristics of meat products. Given the desirable attributes of insects as foodstuff, but low acceptance rate, consumers, who give a high importance to the environmental impacts of food production, are expected to be more likely to consume insects [11,12] and have a higher willingness-to-pay. As opposed to this, insect-based snacks appeared to be least popular from a sensory perspective in a study by de Boer et al. [22]. Based on these findings, we hypothesize that meat consumption habits are important to determine consumers' readiness to consume insect-based products, specifically frequent meat consumers are less likely to switch from meat products to insects.

Consumers perceive insects as healthy due to their nutritional value and high concentration of omega 3 fatty acids [23,24], but they also associate it with a risk of disease and illness as a result of low food safety standards [20,25]. Thus, it is expected that consumers who place increased importance on health aspects and better education are more likely to consume insect-based products. Moreover, the willingness to consume unprocessed insects is substantially lower as compared to the willingness

to consume processed insect products, such as cricket flour [20]. In line with this, de-Magistris et al. [23] find that the willingness-to-pay for bugs significantly reduces when the insects were visualized on the product. Due to these observations, in this study, we offer insect fortified burgers, without visual difference to conventional burgers, to the respondents.

Besides knowing the profile of the group of insect adopters, food manufacturers and traders require specific information on how to set prices and design communication policies. However, only few studies, namely de-Magistris et al. [23] for the Netherlands and Alemu et al. [25] for Kenya, provide specific estimates on the consumer willingness-to-pay for specific insect-based products as a whole and their attributes. The results of de-Magistris et al. [23] reveal that consumers are willing to pay a positive price premium for a box of insect-based sushi equipped with a voluntary logo and a mandatory health claim. Similarly, Kenyan consumers appear to have a positive and significant willingness-to-pay for nutritional value and food safety control attributes for fried termites. The present research paper closes this gap with respect to German consumers. Different to these studies, we allow the respondents to choose between an insect fortified burger and a conventional burger. Hence, we are able to provide concrete willingness-to-pay estimates for one insect-based product in comparison to the conventional meat based product. We expect that the attitudinal characteristics of the respondents are the main driver of their willingness-to-pay for the insect fortified burger. In this study, we estimate a latent class model to link the (observed) choice behavior to socioeconomic and attitudinal characteristics. The empirical model specifically accounts for any heterogeneity in choice behavior by providing group specific willingness-to-pay estimates.

The remainder of the paper is organized as follows. The material and methods section includes a description of the methodology, an overview of the data, and presents the empirical modeling approach. The empirical results of the latent class model and the logit model are reported in section three. The discussion and concluding remarks provide a summary of the main findings and discusses the implications of the study results.

## 2. Material and Methods

### 2.1. Experimental Design

Considering the prevailing disgust of Western consumers towards unprocessed insects [20], we chose burger patties made of minced meat from different sources as the product in the choice experiment. The insect burger patty was made of beef and pulverized insects, which are stirred into the meat mass. We impose that the consumer could not distinguish between the burger made of beef and the burger made of beef and insects by visual inspection, but only by the characteristic of the attribute. An explanation about the production procedure was given to the participants prior to the choice experiment.

In detail, the participants of the choice experiment were asked to choose between two different burger alternatives and a no choice alternative. An exemplary choice set is shown in Figure 1. Each alternative was characterized by six attributes: Production method (conventional vs. organic), health claim, visual impression, carbon emissions, composition of the burger patty, and the purchase price. Purchase price and the level of carbon emission have three levels, while all the other attributes are binary. Each attribute was explained in greater detail to the respondents prior to the experiment. The attributes were chosen based on the significance in the purchase decision of consumers and the characteristics associated to insect based products. The production method and the visual absorption are important attributes of meat products [21]. Both health claims and carbon emissions are related to the source of animal protein as discussed in the introduction. Furthermore, the coefficient estimate for the purchase price facilitates the computation of willingness-to-pay estimates.

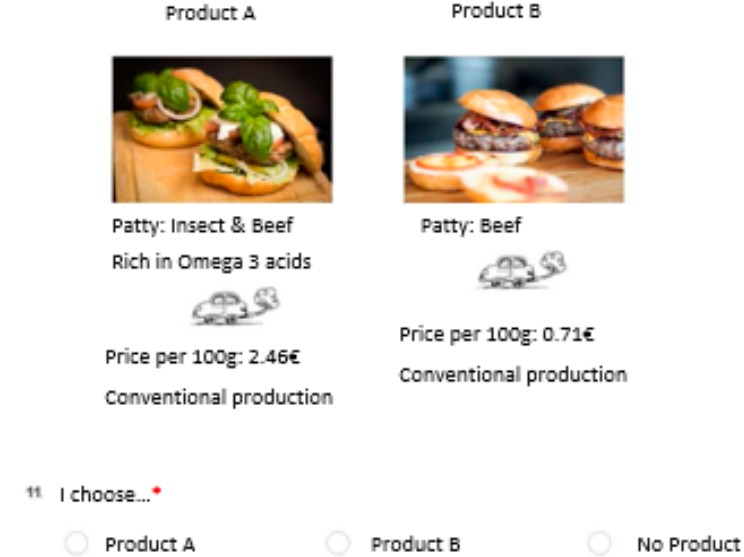

**Figure 1.** Example of the choice set in the experiment.

The different price levels (0.71€, 1.24€, and 2.46€) were chosen based on a survey of existing market prices in supermarkets in Germany. Given that the insect burger of Bugfoundation is sold at 5.99€ for two patties (a 98 g), the prices chosen may seem to be at the lower end. Yet, there is significant room for a reduction in retail prices of the insect burger through the adoption of more cost-efficient production methods and economies of scale in the near future. Notably, it is possible that consumers are offered the insect based burger patty with high emissions or the beef burger patty with low emissions. Certainly, both options are unrealistic. Yet, applying this design is the only way to successfully differentiate between the consumer's willingness-to-pay for the attributes: Environmental sustainability and the composition of the burger patty. Otherwise, no distinction between the preferences for insect burger patties and low $CO_2$ emissions can be made, which will lead to biased estimates of the consumer's willingness-to-pay for insect based food products.

The attributes and their respective levels can be combined to create a number of different choice sets. The full factorial design, which includes all possible combinations of attribute levels, would lead to 144 ($3 \times 3 \times 2 \times 2 \times 2 \times 2$) possible products. Subsequently, the number of choice sets was reduced to a practical number using an orthogonal design, implemented in SPSS, to identify the main effects [26]. Finally, nine choice sets were identified, which consisted each of two burger patty alternatives as well as the no choice option. Each respondent was asked to choose between the alternatives in all nine choice sets.

Prior to responding to the choice decisions in the experiment, consumers were separated into three different groups by letting them choose between a red, white, and black square. As a consequence of the individual decision, a statement was presented to the respondent whenever the respondent chose the red or the black square. The white square was not associated to any prior information. The statement presented after choosing the red square contained information that aimed at encouraging the consumption of insects by pointing at the negative consequence of meat consumption on the global climate, individual health, and conditions in developing countries. In addition, a positive reference about the possible flavor was made. The statement after the black square was neutral only stating that eating insects is harmless to consumers and is widespread in large parts of the world.

*2.2. Data*

The analysis in this study is based on data from a nationwide online survey, which was online from 10–29 of November 2016, conducted by the authors in Germany. The respondents were contacted via social media and email, but mostly came from the northern part of Germany. Out of the 345 respondents

participated in the survey, we could make use of 311 questionnaires. The sample is not nationally representative and mostly includes younger internet enthusiastic consumers. Out of the 311 valid respondents, 50 were vegetarians who were excluded from the choice experiments. The participants knew that the survey was about novel foodstuffs, but they were not aware about the type of product offered in the choice experiment. The large number of vegetarians confirms that there was relatively little selection bias. The average respondent was predominantly female (73.2%) and 30.85 years old. While a nationally representative sample has several advantages, our study was targeted at younger consumers, who are found to be more likely to adopt insects into their diet in related studies [10,12]. Given that edible insects are not yet accepted by the large majority of consumers, we decided to focus on the group of possible early adopters.

Alongside the choice experiment, respondents were asked to also respond to several food consumption related statements and to state their socioeconomic status. A full description of the variables and their respective means are presented in Table 1. The selection of individual specific variables employed in the analysis was informed by a cluster analysis and the findings of existing studies on the subject by Verbeke [12] and Hartmann et al. [19], who both link the readiness to consume insects in the future to attitudinal and socioeconomic variables. Following these studies, we also examine the determinants of future readiness for insect consumption as well as past experience. In doing so, we use the dummy variable, future readiness, which is equal to 1 if the respondent considers eating insects in the future as well as the variable of past consumption, which is equal to 1 if the respondent has eaten insects before. Within our sample, about 26% of the respondents reported that they have eaten insects before and 39% of the sample states planning to do so in the future. Interestingly, almost a third of the respondents, who had tried insects before, does not plan to eat them in the future. Both past experience and future readiness in our sample are substantially higher than in the sample of Hartmann et al. [19].

**Table 1.** Membership variables and description.

| Variable Name | Description | Sample Mean |
| --- | --- | --- |
| Aversion Insects | Aversion towards eating insects (7-point scale) | 2.30 |
| Disgust | Level of disgust towards insects (7-point scale) | 4.07 |
| Future readiness | 1: Ready to eat insects in future, 0: Otherwise | 0.39 |
| Past consumption | 1: Ate insects before, 0: Otherwise | 0.26 |
| Food Trend | Preferences for learning about food products and recipes (7-point scale) | 3.19 |
| Food neophobia | Aversion towards eating unfamiliar food products (7-point scale) | 4.50 |
| Attitude Bio | Attitude towards organic production method (7-point scale) | 4.83 |
| $CO_2$ Emissions | Attention to the $CO_2$ emissions in production method (7-point scale) | 3.01 |
| Convenience | Preference for convenience food (7-point scale) | 2.91 |
| Nutritional information | Importance of nutritional information on food products (7-point scale) | 4.57 |
| Taste | Importance of taste of food products (7-point scale) | 6.40 |
| Health statement | Importance of health statement on food products (7-point scale) | 3.10 |
| Gender | 0: Female 1: Male | 0.27 |
| Age | Age in years | 30.46 |
| Low Education | 1: If no level received, 0: Otherwise | 0.10 |
| Education completed | 1: If education was completed, 0: Still in training | 0.62 |
| Meat frequency | Frequency of weekly meat consumption (1: More than three times 2: 1–3 times 3: Less than one time.) | 1.96 |
| Positive statement | Respondent received positive statement | 0.38 |
| No statement | Respondent received no statement | 0.40 |

Food neophobia and disgust towards insects were identified as major obstacles to the adoption of insects into human diets. Food neophobia was measured using a slightly modified version of the scale developed by Pliner and Hobden [27] due to missing observations for a significant share of the sample. The respective list of questions is reported in Table A1 in the Appendix A. In addition to that, we also include a variable (aversion insects) that also explicitly captures the respondents' attitudes towards insects. The variable food trend describes the consumer preference for street food festivals, food blogs, and the attendance of cooking classes, which is also included to factor in the respondent's preference

to adopt alternative consumption habits. Attitude bio reveals the preference of the respondents for an organic and natural production method.

Moreover, the variables of $CO_2$ emissions, convenience, nutritional information, health statement, and taste show consumers' attitudes regarding main characteristics of meat products. Last, the frequency of meat measures the extent of meat consumption of the respondents. As the socio-economic variables, we include the age in years, gender, education (a dummy variable for a low education level), and as an indicator for income, a dummy variable of whether the respondent has completed her educational training. Income is omitted from the regression because a significant number of respondents refused to declare their income. Last, we also test whether the provision of specific statements, as discussed above, has an impact on the readiness to consume insects in the future.

### 2.3. Statistical Modelling

Stated choice experiments are based on the characteristic theory by Lancaster, according to which, the utility of consumption, $U$, is derived through individual attributes, rather than by the intrinsic value of a consumer good [28]. However, in a choice situation, real or hypothetical, consumers may not always act perfectly rational and determined, which introduces a random error component into the model [29]. The preferences of the consumer, $n$, cannot be observed in reality, but the choice for a specific product alternative with specific attributes can. The multinomial decision in the choice experiment can be rewritten as a binary decision problem, where the choice indicator, $C_{nj}$, describes whether the consumer, $n$, chooses an alternative, $j$, of a given choice set or not:

$$C_{nj} = f(U_{nj}) = \begin{cases} 1 \text{ if } U_{nj} = \max_j\{U_{nj}\} \\ 0 \text{ otherwise} \end{cases} \tag{1}$$

The consistent estimation of the preference parameters, $\beta$, requires that the error terms are independently and identically distributed. This assumption is very restrictive when participants of the choice experiment make multiple choices. The conditional logit model relaxes this assumption, but still assumes homogeneous preferences across all individuals. However, it is widely recognized that consumers are heterogeneous in their taste and preferences [30,31]. The conditional logit latent class model (LCM), on the other hand, allows simultaneous classification of consumers into homogeneous groups and explains the choices of these consumer groups which makes it preferable to the mixed logit model [32].

In the LCM, consumer heterogeneity is explained by explanatory variables, such as individual characteristics as well as consumer attitudes and perceptions. The covariates are used to estimate the probability of class membership for the participants of the choice experiment. Membership is an unobserved latent variable and the likelihood function of the membership, $M^*_n$, is given by:

$$M^*_n = \lambda Z_n + \varphi_n \tag{2}$$

where $Z_n$ is a vector of the determinants of class membership and $\varphi_n$ is an error term, which is assumed to be independently and identically distributed across individuals and consumer segments with a type I extreme value distribution [33]. Thus, the unconditional choice probability of an individual, $n$, choosing the alternative, $j$, in a choice set, $k$, can be expressed as:

$$P_{jkn} = \sum_{c=1}^{C} \left[ \frac{\exp(\vartheta\gamma^c Z_n)}{\sum_{c=1}^{C} \exp(\vartheta\gamma^c Z_n)} \right] \left[ \frac{\exp(\alpha_j^C + \beta_j^C X_{jkn} + \sum_A \gamma_{Aj}^C A_{jkn})}{\sum_{i \neq j} (\alpha_i^C + \beta^C X_{ikn} + \sum_A \gamma_{Ai}^C A_{ikn})} \right] \tag{3}$$

where j = 1,..., J are alternative food products; k = 1,..., K are the choice sets provided to the respondents; c represents the latent classes; and A is the attribute of each alternative. The parameter, $\beta^C$, is the class specific price effect and $\gamma_A^C$ is the class specific effect of attribute A. Using the price as an attribute

allows the willingness-to-pay for the attributes included in the choice set to be computed. In our case, we expect consumers to accept insect-based burger patties only when receiving a price discount. In this case, the WTP is negative and can be interpreted as the willingness-to-accept an inferior product attribute. Given that the utility is a linear function of all attributes, the WTP for the different consumer classes, *C*, can be computed as:

$$WTP_A^C = -\gamma_A^C / \beta^C \tag{4}$$

where $\gamma_A^C$ is the coefficient for the attribute, *A*, in consumer class, *C*.

To make the choice situation more realistic and to reduce overall bias, it is recommended that a no choice option in each choice set is included [34], as illustrated in Figure 1. However, this demands additional caution in the econometric model. As discussed by Haaijer et al. [35], it is not an option to assign zeros for each attribute to the no choice alternative. Instead, it is recommended to include a dummy variable indicating the no choice option and to apply effect coding to differentiate the no choice option from the base attribute level. For instance, the variable bio assumes 1 for organically produced meat, –1 for conventionally produced meat, and 0 for the no choice alternative. The full list of attributes and their respective level, as they are used in the model, are presented in Table 2.

**Table 2.** Attribute levels and variable names used in choice experiment.

| Attribute | Attribute Level | Variable Name |
|---|---|---|
| Price | 1. 0.71 €/100g<br>2. 1.24 €/100g<br>3. 2.48 €/100g | price |
| Production method | 1: Organic, –1: Conventional | bio |
| Patty composition | 1: Beef + insects, –1: Only beef | pat_i |
| Health claim | 1: Yes (contains omega 3 fatty acids), –1: No | hc |
| Visual impression | 1: Healthy, –1: Unhealthy | pic_h |
| $CO_2$ Emissions | 1: One car (low emissions), 0: Two cars (medium emissions),<br>–1: Three cars (lots of emissions) | co2_1 |
| | 1: Two cars (medium emissions), 0: One car (low emissions),<br>–1: Three cars (lots of emissions) | co2_2 |

Using this approach, the dummy for the no choice alternative represents the utility from not choosing any of the products. A positive value for the no choice dummy indicates positive utility from not choosing any of the products. A negative value is equivalent to a positive utility for choosing any of the burger patties. Furthermore, following the proposal of Haaijer et al. [35], we mean center the linear attribute prices. As an indirect result of effect coding, the utility from each attribute is compared to the no choice alternative instead of a base attribute. In consequence, the interpretation of the willingness-to-pay estimates needs to be adjusted.

In this case, the utility derived from each attribute is compared to the no choice alternative instead of the base attribute, which alters the formula used to compute the WTP. Allowing also the possibility of effect coding, the class specific WTP for attribute A is as follows [36]:

$$WTP_A^C = -1/\beta^C (\gamma_A^C \Lambda_A - \gamma_{NC}^C) \tag{5}$$

where $\gamma_{NC}^C$ is the class specific estimate for the no choice option and $\Lambda_A$ represents a categorical variable representing the value of the specific characteristic of the attribute, A. Using effect coding, the no choice alternative takes the value, zero, and for the other characteristics, the values of 1 for yes and –1 for no. Thus, the willingness-to-pay for a characteristic of an attribute, for instance, the organic production method in comparison to the conventional production method doubles the value obtained by Equation (4). The results of the willingness-to-pay estimate, according to (5), are reported in Table A2 in Appendix A. These willingness-to-pay estimates express the amount consumers would spent for a burger patty with the respective attribute in comparison to not choosing any product.

Before the analysis of the choice experiment, we also run ordinary logit regression to identify socio-economic characteristics and attitudinal variables as possible sources of the consumer heterogeneity. This enables us to determine the variables used in the membership equation of the LCM (although feasible, we do not use the contingent valuation method for the WFP estimation). The structure of the regression model is as follows:

$$D_n = \kappa Z_n + e_n \qquad (6)$$

where $D_n$ is a binary indicator (i) taking the value of 1 if the respondent had experience eating insects, and zero otherwise, and (ii) taking the value of 1 if the respondent plans to eat insects in the future, and zero otherwise. $\kappa$ is a vector of the parameters for the explanatory variables, $Z$.

## 3. Results

### 3.1. Binary Choice Models

In this section, we discuss the results of the two binary choice models and the LCM. We begin by looking at the determinants of past insect consumption and future readiness for insect consumption, which we use to determine the membership variables for the LCM in the subsequent analysis. Table 3 presents the coefficient estimates, the estimated marginal effects for the binary choice models, and the respective Wald statistics in brackets. Overall, according to the pseudo R squared, the attitudinal and socio-economic variables selected explain about a third of the variation in future readiness, but less than 15% of the variation in past consumption.

The variables, taste, food trend, attitude bio, nutrition information, age, and education completed, are not found to have significant effects on either future readiness or past consumption of insects. An increase in the aversion towards new food products, measured by the food neophobia score, by one point in the Likert scale reduces the likelihood of future readiness by 7%. Similarly, a one point increase in the respondent's level of disgust towards insects is associated with a decline of the likelihood of future insect consumption by 11.2%. Disgust also plays an important role in explaining past insect consumption, as indicated by a significant negative coefficient estimate, which is equivalent to a decrease in the likelihood of past consumption by 6% for an increase in disgust by one point. The general aversion towards insects is negatively related to the likelihood of future readiness. Precisely, a one point higher attitude towards insects score is associated with a decrease in the likelihood of future readiness by 17%.

The respondents, who are interested in consuming climate-friendly products, are significantly more likely to be willing to adopt insect consumption. An increase of the related score ($CO_2$ emissions) by one point is associated with a 6% higher likelihood to consume insects in the future. In the same vein, the respondents who report infrequent meat consumption (only once a week) are significantly more likely to declare their future willingness to consume insects. A one point lower frequency of meat consumption score increases the consumer's willingness to consume insects in the future by 13.9%.

The other variables that describe the importance of product characteristics, such as organic production method and nutritional information, do not appear as significant drivers of future readiness and past consumption of insects. Only the coefficient of the variable, health statement score, is significantly different from zero, but only in the regression for past insect consumption. Respondents with a one point greater concern for the health statement associated with the food product are 6% more likely to have consumed insects in the past.

Finally, among the socio-economic variables, only gender and low education are found to have a significant impact on future readiness, while none of the coefficient estimates are significantly different from zero in the regression explaining the decision for past insect consumption. According to the regression results, male respondents are 42% more likely to express interest in future insect consumption. A low education level of the respondent is associated with an 18% lower likelihood of future readiness, however, the coefficient is only moderately significantly different from zero (10%).

**Table 3.** Binary choice logit model: ML estimates.

|  | Future Readiness | dy/dx | Past Consumption | dy/dx |
|---|---|---|---|---|
| Food neophobia | −0.366 ** | −0.0782 | −0.195 | −0.0423 |
|  | (−2.14) |  | (−1.27) |  |
| Taste | 0.26 | 0.0554 | −0.143 | 0.0035 |
|  | (1.08) |  | (−0.51) |  |
| $CO_2$ Emissions | 0.281 ** | 0.0601 | −0.0816 | −0.0130 |
|  | (1.99) |  | (−0.66) |  |
| Disgust | −0.522 *** | −0.1116 | −0.416 *** | −0.0656 |
|  | (−4.26) |  | (−3.87) |  |
| Food trend | 0.0476 | 0.0101 | 0.164 | 0.0193 |
|  | (0.37) |  | −1.25 |  |
| Aversion insects | −0.797 *** | −0.1704 | (0.164) | −0.0311 |
|  | (−3.86) |  | (−0.94) |  |
| Attitude bio | −0.0104 | −0.0022 | 0.0283 | −0.0060 |
|  | (−0.06) |  | (0.16) |  |
| Convenience | 0.588 * | 0.1257 | −0.338 | −0.0385 |
|  | (1.81) |  | (−1.12) |  |
| Gender | 1.869 *** | 0.4228 | −0.201 | −0.0459 |
|  | (3.95) |  | (−0.47) |  |
| Nutrition information | 0.0525 | 0.0112 | −0.0507 | −0.0098 |
|  | (0.42) |  | (−0.49) |  |
| Health statement | 0.206 | 0.0440 | 0.298 *** | 0.0618 |
|  | (1.59) |  | (2.72) |  |
| Age | −0.0272 | −0.0058 | −0.00319 | −0.0008 |
|  | (−1.47) |  | (-0.19) |  |
| Education low | −1.009 * | −0.1786 | −0.665 | −0.1214 |
|  | (−1.78) |  | (−0.91) |  |
| Education completed | −0.601 | −0.1312 | −0.368 | −0.0485 |
|  | (−1.61) |  | (−1.10) |  |
| Frequency meat | 0.648 ** | 0.1386 | −0.167 | −0.0077 |
|  | (2.4) |  | (−0.67) |  |
| Positive information | 0.695 | 0.1520 |  |  |
|  | (1.47) |  |  |  |
| No information | 0.307 | 0.0662 |  |  |
|  | (0.66) |  |  |  |
| _cons | −1.615 |  | 3.089 |  |
|  | (−0.72) |  | (1.24) |  |
| Observations | 244 |  | 246 |  |
| R^2 | 0.3436 |  | 0.1338 |  |

z-statistics in parentheses; * $p < 0.10$, ** $p < 0.05$, *** $p < 0.01$; dy/dx is marginal effect.

### 3.2. Predicted Probabilities

To illustrate the implications of the attitudinal characteristics of the respondents, we present the predicted probabilities of being ready to consume insects in the future and the predicted probabilities of past consumption of insects in Figures 2–4 for selected attitudinal variables of the respondents and across the range of the values of the respective variable. Figure 2 shows the probability of being ready to consume insects in the future dependent on the consumers' aversion towards insects and the food neophobia score. Both variables are measured by a score from 1 to 7. The predicted probabilities indicate how different levels of the variables are related to the likelihood of future insect consumption holding all other variables at the means. For instance, a respondent with a food neophobia score of 1, which is the lowest level of reluctance to eat new food products, has a probability of almost 80% to be ready to consume insects in the future. On the other hand, respondents with a food neophobia score of 7 are ready to consume insects in the future with a probability of only 2%. Generally, the association is linear. The importance of the aversion towards insects score on the predicted probabilities appears to be less pronounced and has a negative exponential trend. At the sample mean of 2.3, a respondent's probability of being willing to consume insects in the future is about 35% and increases to above 60% for a score of 1, while it remains constantly low at a score above 4.0.

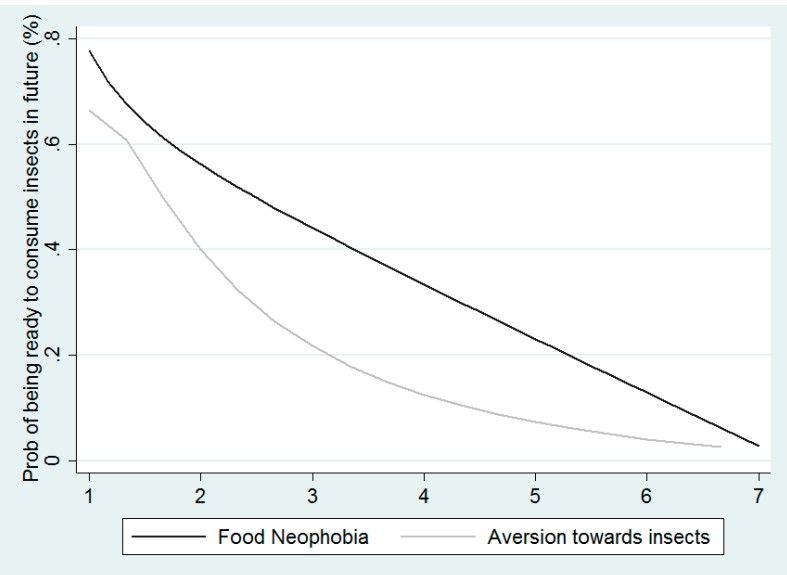

**Figure 2.** Predicted probability of being ready to consume insects in the future on the food neophobia and the aversion towards insects scores.

In a similar vein, we illustrate the predicted probabilities for two additional attitudinal variables, namely the importance of $CO_2$ emissions (7 point Likert scale) and the interest in convenience products (4 point Likert scale) scores in Figure 3. In both cases, a higher score is associated with a greater likelihood of being ready to consume insects in the future. For instance, when all other variables are at their means, a 5.0 score for the importance of $CO_2$ emissions relates to a probability of being ready to consume insects in future of 54%. The progression is linear. Against this, the predicted probabilities for the importance of the convenience score is constant up to a score of 2.0 and increases exponentially thereafter up to a predicted probability of 55%.

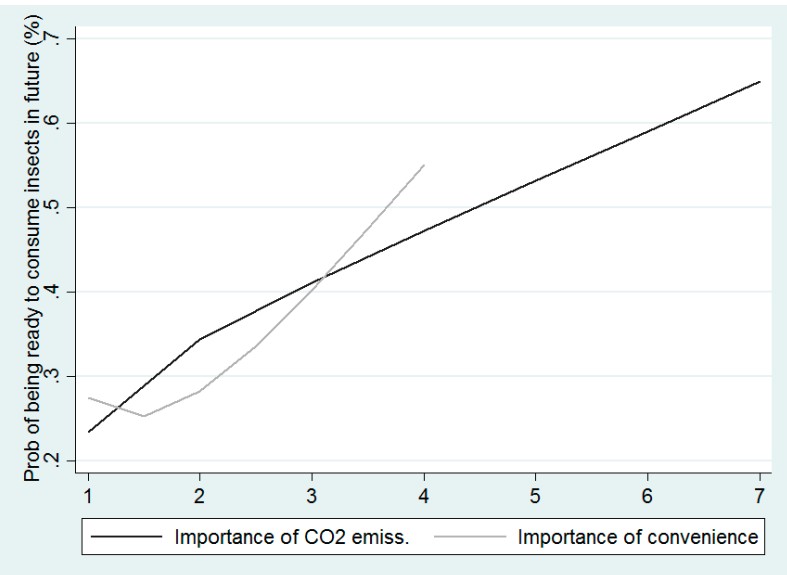

**Figure 3.** Predicted probability of being ready to consume insects in the future on the importance of $CO_2$ emissions and the importance of convenience products scores.

Last, we present the predicted probabilities of past insect consumption dependent on the scores of disgust and the importance of the health statement in Figure 4. The disgust score is negatively associated with the probability of past insect consumption and varies between 52% (score of 1.0 for no

disgust) and 10% (7.0 score of high disgust). The distribution of the disgust score is relatively equal with about 10% of the respondents having a score of 1.0 and 13% having a score of 7.0. The predicted probabilities of past insect consumption on the score of the importance of the health statement vary only between 23% and 37%. At the sample mean of 3.1, the predicted probability is around 27%. This complies with the profile of about 20% of the respondents, while a score greater than 5.0 is the rare exemption.

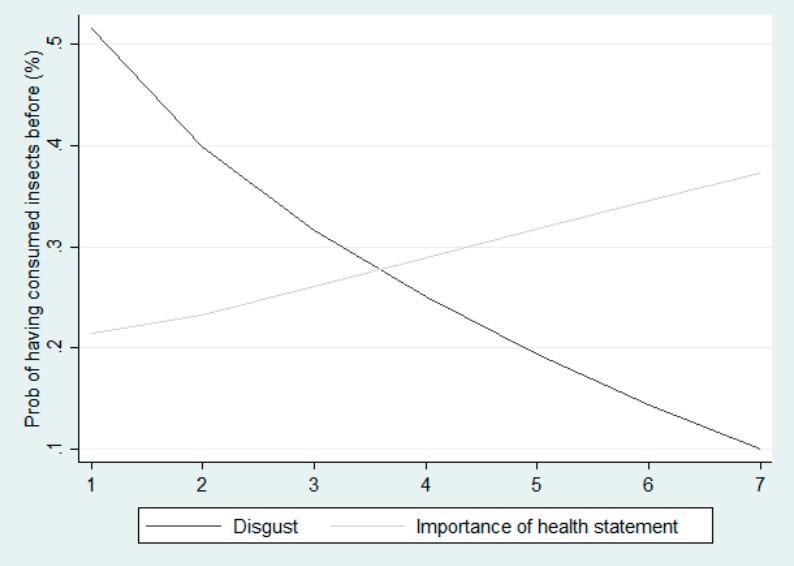

**Figure 4.** Predicted probability of past insect consumption on disgust and the importance of health statement scores.

### 3.3. Latent Class Model

In order to reduce the variables used in the membership equation of the LCM to a practical number, we include the variables of 'food neophobia', '$CO_2$ emissions', 'disgust', 'attitude insects', 'convenience', 'gender', 'education low', and 'frequency meat', which appeared to be significant, at least, at the 10% level of significance in the binary choice regressions, in the membership equation of the LCM. In addition, the attitude towards organic products is included, as one of the attributes of the burger patty is the organic production method. The information provided to the respondents prior to the experiment did not have an effect on their willingness to adopt insect consumption in the future. Therefore, the variable associated with the specific information provided was excluded from the subsequent LCM analysis. Similarly, we found that past insect consumption was not significant in explaining class membership, and therefore was excluded from the final model.

The model selection, which is described in detail in the technical appendix, revealed that there are three latent classes in our sample. The coefficient estimates for the individual attributes and the class membership determinants of the maximum likelihood estimator are shown in Table 4. The respective coefficient estimates for class 1, class 2, and class 3 indicate choice heterogeneity of the participants, which can be seen by differences in the magnitude and significance of the coefficient estimates for the attributes. In addition, the coefficient estimate for the determinants of the class membership suggest that this heterogeneity is driven by attitudinal and socioeconomic characteristics of the respondents. The magnitude of the coefficient cannot be interpreted easily, but is usually put in relation to the coefficient estimate for the attribute price. As illustrated above in Equation (4), the quotient can be interpreted as the willingness-to-pay for the respective attribute. Since the classes are latent, it is not possible to compute marginal effects on the basis of the coefficient estimates for the individual membership classes.

**Table 4.** Three latent class conditional logit model: ML estimates.

| | Class 1 | Class 2 | Class 3 |
|---|---|---|---|
| *Attributes* | | | |
| price | −0.3170 *** | 0.1981 | −0.6842 **** |
| | (0.1387) | (0.2574) | (0.0795) |
| bio | −0.0254 | 0.7398 *** | 0.7696 *** |
| | (0.1191) | (0.1599) | (0.0832) |
| pic_h | 0.3411 ** | 0.1563 | 0.01977 |
| | (0.1445) | (0.1375) | (0.1083) |
| hc | 0.0954 | −0.0702 | 0.3876 *** |
| | (0.1102) | (0.1192) | (0.0625) |
| pat_i | −1.6290 *** | −0.6496 *** | −0.1523 * |
| | (0.1420) | (0.1825) | (0.0788) |
| co2_2 | −0.1862 | −0.1966 | 0.3270 ** |
| | (0.1921) | (0.2417) | (0.1395) |
| co2_1 | 0.5280 *** | 0.8152 *** | 0.6795 *** |
| | (0.1417) | (0.2343) | (0.0956) |
| none | −0.8506 *** | 1.8989 *** | −3.1118 *** |
| | (0.1799) | (0.2598) | (0.3521) |
| *Membership variables* | | | |
| Food neophobia | 0.3043 ** | 0.3571 | |
| | (0.1530) | (0.2283) | |
| $CO_2$ Emissions | −0.2204 | −0.3026 * | |
| | (0.1354) | (0.1675) | |
| Disgust | 0.5839 *** | 0.1263 | |
| | (0.1273) | (0.1615) | |
| Aversion insects | 0.3799 ** | 0.0814 | |
| | (0.1829) | (0.2464) | |
| Attitude bio | −0.1874 | 0.7041 *** | |
| | (0.1773) | (0.2555) | |
| Convenience | 0.1981 | 0.2614 | |
| | (0.2902) | (0.4243) | |
| Gender | −0.0550 | 0.1211 | |
| | (0.4698) | (0.6927) | |
| Health statement | −0.2547 ** | -0.1070 | |
| | (0.1296) | (0.1522) | |
| Education low | 0.6340 | 1.7299 ** | |
| | (0.8136) | (0.8237) | |
| Frequency meat | −0.9410 *** | 0.0254 | |
| | (0.2999) | (0.3571) | |
| cons | −1.1772 | −6.4169 *** | |
| | (1.1826) | (2.2948) | |
| Latent class probability | 0.367 | 0.155 | 0.478 |
| Number of choice sets | 2214 | | |

Standard errors in parentheses; * $p < 0.10$, ** $p < 0.05$, *** $p < 0.01$.

The no choice option has a significant impact on the probability to choose an alternative for all three classes. Respondents of class 1 and class 3 are found to receive negative utility from choosing none of the products. By contrast, respondents of class 2 seem to prefer the no choice option over consuming any type of the burger patties, showing their distaste for the product. Both class 1 and class 3 respondents have a significant negative parameter estimate for the attribute prices, which is in accordance with the economic theory. The parameters' estimate for class 3 is twice as large as the estimate for class 1, indicating a greater price sensitivity for burger patties of respondents in this class. Again, respondents of class 2 are different to the other two classes, as shown by an insignificant coefficient estimate for the attribute price.

As expected, the insect beef mixed burger patty, instead of the pure beef patty, does not create additional utility to the respondents. Instead, members of class 1 and class 2 show a significant dislike

towards the attribute. For respondents of class 3, the parameter estimate is only significant at the 90% level of significance and the point estimate is also substantially below the estimate for class 1 and class 2, which means that they may be willing to purchase the insect based burger patty if the other attributes, for instance, the price, do compensate for the loss in utility. The density distribution of the coefficient estimate for the insect based burger patty is displayed in Figure 5. The attribute organic production method has a significant coefficient estimate only for members of class 2 and class 3, while the magnitude of the coefficient estimate suggests that these respondents are quite keen on choosing organic meat products. For members of both classes, the picture of the healthy burger does not seem to provide additional utility, as indicated by the insignificant coefficient estimates.

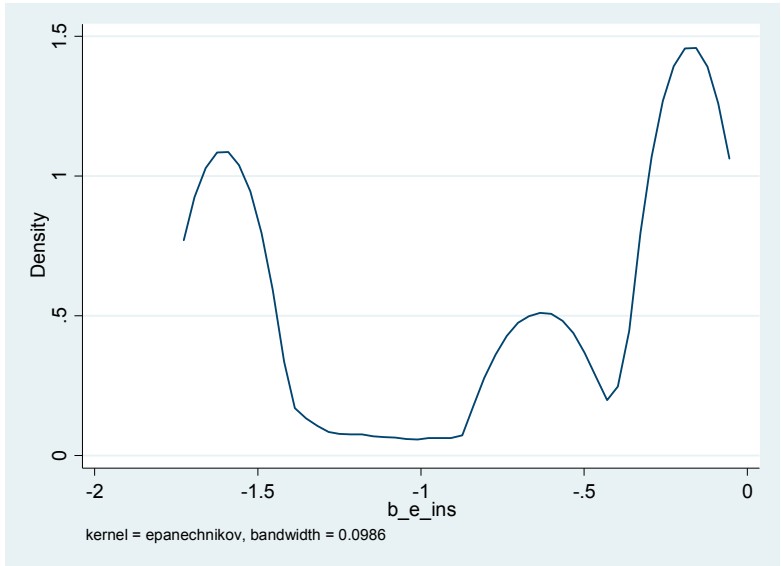

**Figure 5.** Density distribution of the coefficient estimate for the insect patty.

The positive health claim regarding the saturated fatty acids content of the product is associated with a significantly higher probability of choosing the respective product for members of class 3 only. Last, while respondents of all classes are more likely to choose products with low $CO_2$ emissions, only class 3 respondents are also more likely to choose products with a medium amount of $CO_2$ emissions. The coefficient estimates illustrate the utility in comparison to the base category, which is a high amount of carbon dioxide emissions.

Overall, the model predicts that almost half of the respondents belong to class 3, while a third were assigned to class 1 and only 15% to class 2. The differences in coefficient estimates for the individual attributes discussed above may be explained by the significance of the class membership variables in Table 4. A positive coefficient estimate indicates a greater probability that the respondent is a member of class 1 or class 2, in comparison to being a member of the base class 3. Conversely, a lower value of the corresponding coefficient estimate shows a higher probability for the respondent's membership to class 3. As an example, the coefficient of the first variable 'food neophobia' is positive for both members of class 1 and class 2, which means members of class 3 are characterized by a significant lower level of aversion towards new food products.

Besides greater food neophobia, higher disgust, and aversion towards insects, a lower importance given to the health statement, and more frequent meat consumption characterize respondents in class 1, in comparison to the members of class 3. Interestingly, a positive attitude towards organic products increases the likelihood of belonging to class 2, as shown by the positive and significant coefficient estimate for the variable attitude bio, but a high preference for low $CO_2$ emissions reduces the likelihood of belonging to class 2. Moreover, a lower educational background increases the likelihood of class 2 membership. Both respondents in class 2 and class 3 are less frequent meat consumers than

respondents classified into class 1. The coefficient estimates of the variables, convenience and gender, appear to be insignificant, suggesting that they are unimportant in determining the class membership for both class 1 and class 2 respondents. In comparison to the other two classes, respondents who belong to class 3 are more interested to try new foodstuffs and prefer products with healthy and resource preserving attributes.

Based on the coefficient estimate of Table 4, we proceed to compute the class specific willingness-to-pay estimate for the individual product attributes. Figure 6 presents the willingness-to-pay for specific attributes in comparison to the base attribute. The latter estimates are commonly reported in willingness-to-pay studies. We present the willingness-to-pay estimate only if both the coefficient estimate for the attribute and the prices was found to be significant. Accordingly, members of class 1 have a large willingness-to-pay of 1.94€ for products with a healthy visual impression and low $CO_2$ emissions (2.76€), but need to be compensated for eating insect burger patties. Indeed, the willingness-to-accept an insect based burger patty is as large as 9.80€, which is not a realistic price. On the other hand, respondents classified as members of class 3 have a positive willingness-to-pay for the attributes of bio and low $CO_2$ emissions. Specifically, they would pay 2.72€ more for organically produced burger patties and 2.11€ for low $CO_2$ emissions (as compared to a high level of emissions). Moreover, they have a moderate willingness-to-accept of 1.13€ for products with a positive health claim. In contrast to the members of class 1, the willingness-to-accept insect based burger patties is not restrictively high. In detail, on average these consumers are indifferent between pure beef and beef insect mixed burger patties if the beef-insect mixed patty was 0.47€ cheaper.

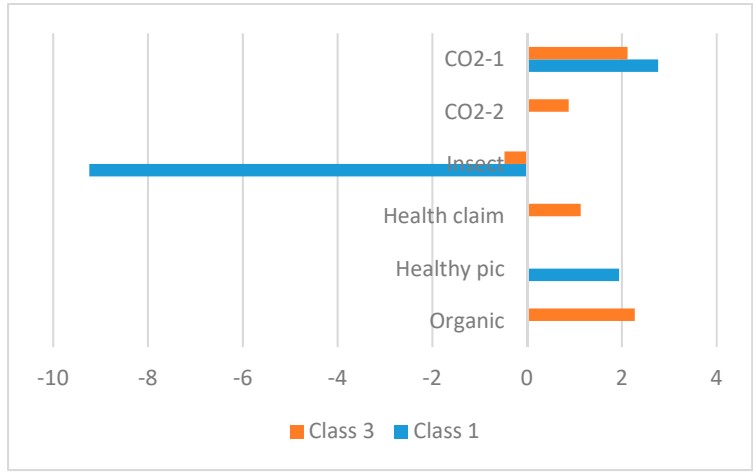

**Figure 6.** Willingness to pay estimates for different attributes.

## 4. Discussions and Conclusions

### 4.1. Discussions

The study at hand examines the readiness of German consumers to eat food products made of insects. In addition to that, we conducted a choice experiment to understand the willingness-to-pay for insect based products in a relatively moderate form, specifically in the form of insect fortified burgers made of a mixture of beef and insects without a distinguishable optic difference. About one quarter of the sample had prior experience with insect consumption and about 40% reported to be willing to eat insects in the future. These findings contrast with other studies, who report lower figures on the readiness of consumers for insect consumption in Western countries [11,12,19].

This could have multiple explanations. First, our sample was drawn from an online survey. Online surveys usually comprise of younger and more informed respondents and are less representative of the whole population. Thus, we need to compare our findings with those of studies with a similar sample composition, such as Caparros Megido et al. [37], Ruby et al. [18], and Wilkinson et al. [38],

who also surveyed only a specific group of respondents. For instance, more than three quarters of the respondents in Caparros Megido et al. and Ruby et al. [18,37] claimed to be interested to eat insects in the future, while 21% of the respondents in the study of Wilkinson et al. [38] reported previous experiences with insect consumption. Second, our study took place a couple of years after the earlier studies. The progressing time might have had a positive effect on the consumer acceptance and social acceptability of eating insects, which had to do with the media presence of the topic and the publication of a number of positive studies on the topic. Last, the question on future readiness to consume insects was posed after the choice experiment with the insect fortified burger, which could have influenced the answer of the respondents. Most consumers usually think about unprocessed mealworms and larvae when they are offered insects to try, and the prospect of eating an insect burger might have positively shaped their views. This is in line with the findings by Hartmann et al. [19], who report a higher readiness of German consumers for insects in a processed form, as compared to unprocessed insects.

To shed light on the research questions raised in this study, we can draw on our findings from both the binary choice model and the choice experiment. The choice experiment allowed the testing of whether the explicitly stated readiness to consume insects in the future is synonymous to the implicit readiness revealed by the product choice in the experiment. We find that the readiness of consumers to adopt insects into their diet is strongly related to attitudinal variables. The possible early adopters are more concerned about the environmental and health related consequences of their consumption behavior. They also seem to consume meat products less often than consumers who are not ready to eat insects. As opposed to this, the group of consumers who is not (yet) ready to consume insects is characterized by greater levels of disgust, a higher food neophobia score, and a higher general aversion towards insects. The importance of these variables is in line with the findings of Verbeke [12], who attribute a large part of the limited adoption of insects into human diets by food neophobia. Similarly, Hartman et al. [19] concluded that food neophobia and disgust play a significant role for German consumers' willingness to eat insects.

Combining the coefficient estimates and the determinants of the class membership is very meaningful and let us describe the consumer group of early adopters of insect-based food products and their attitudes. Therefore, we classify members of class 3 as "adventurers and ecological consumers", while class 1 contains "insect neophobic conservative consumers". Class 2 respondents seem to be ecologically concerned and infrequent meat consumers and are unlikely to belong to the market segment for burger patties, regardless of the insect content. Respondents of the class of "adventurers and ecological consumers" also have positive and significant willingness-to-pay for organically produced burger patties, a positive health claim, and lower levels of $CO_2$ emissions during the production of the burger meat. This indicates that a higher preference for healthier products is a main motivation to adopt insects into human consumption as a substitute for meat products, which are often, if frequently consumed, considered to increase the risk of cardiovascular diseases. Insect consumption, on the other hand, is not associated with an increasing health risk, and some research even provides evidence for bioactive compounds in insects, which could have the potential to reduce health risks [1]. In line with this, potential adopters of insects appear to be better educated than non-adopters. The remaining respondents are not willing to consume insect-based products in the near future. However, they also reveal a positive willingness-to-pay for a healthy visual impression of the burger and lower $CO_2$ emissions. The class of "adventurers and ecological consumers" is more interested in trying new and exotic foodstuffs, which supports the findings by Hartmann and Siegrist [39].

The willingness-to-pay estimates are generally high in comparison to the price levels chosen in the experiment. We believe that this does not challenge the results of the estimation, it rather hints at possible scale effects caused by the picture presented to the consumers in the choice experiment. The picture showed a full burger, which could be served in a restaurant, instead of the pure burger patty. The information statement, which was presented conditional on the choice of the respondents, had no significant impact on neither class membership nor the coefficient estimates of the attributes.

*4.2. Limitations and Future Research*

The small and unrepresentative sample does not allow conclusions on the size of the market for insect-based food products. The sampling focused on a specific group of consumers, namely younger better educated consumers, who have been identified as the group of early adopters by other studies. In this respect, our sample is, intentionally, quite different to the samples of related works. Besides the limitations of the sample and possible shortcomings in the implementation of the experiment, the LCM model also does not account for two methodological issues present in choice experiments. On the one hand, the choice situation is purely hypothetical for the respondents. On the other hand, scale heterogeneity exists, which was identified as an important driver of preference heterogeneity. It implies that the idiosyncratic error is different for some respondents than for others, which could alter the coefficient estimates. To generate better information for the food industry with respect to the market potential of insects as meat substitutes, a larger and more representative sample is needed and more advanced econometric techniques should be used. For this purpose, the use of other sampling techniques should be considered, such as face-to-face interviews in supermarkets or online surveys with stratification. Nevertheless, future studies can benefit from the findings of this study and examine the issues outlined here in more detail.

The stated preference methods are generally more appropriate as long as products are not available in the market. Recently, the availability of insect-based food products, ranging from protein powder over pasta to the insect fortified burger of Bugfoundation has increased substantially. Therefore, future studies should analyze market data to better understand the antecedents of consumer behavior. This includes the probability of regular consumption as distinguished from onetime purchases driven by curiosity. Moreover, the attribute taste has not been specified in detail in our study. Instead, it was left to the respondents to make assumptions about the possible taste of insect-based products. Future studies need to analyze the sensory characteristics of different types of insect-based food products and associate it with individual preferences and consumer behavior.

*4.3. Conclusions and Implications*

The present study attempts to provide information about consumer behavior with respect to insect consumption in Germany. The choice experiment reveals the first willingness-to-pay estimates for insect based products in Germany. The findings enable food processing companies to profile the consumer segment of early adopters and to improve communication strategies based on the attitudinal variables that characterize consumers in this segment. The willingness-to-pay estimates also aid in the pricing of insect-based food products in Germany.

In general, the participants of the survey were found to be very diverse, which also resembles the current situation in Germany. Revisiting our research question raised in the title, we conclude that some consumers feel disgust when thinking about eating insects, but others see edible insects as an innovative solution to the adverse effects of the Western life style. The latter group of consumers, about 45% of our sample, are characterized as "adventurers and ecological consumers", who are willing to eat the insect-based product without a substantial price reduction, while the other respondents are unlikely to consume insects in the future. This result hints at the possibility that there may be a market for insect based food products as a meat substitute in Germany. However, a general conclusion would require the obtainment of a better understanding on the determinants of regular insect consumption instead of the one-time decision to opt for an insect fortified burger in the choice experiment. Disgust and aversion towards insects was identified as the main obstacle to wider adoption of insects into human consumption (see also [39]). One option would be to produce insect based products that are very similar in taste and appearance to conventional meat products. Second, the current price of around 3 €/100 g is much higher than what consumers are willing to spend. Given the fact that insect cultivation is often still manual, there is a large potential to improve production processes. To achieve commercial large-scale production, current farming systems need to include automation of some key processes to increase economic competitiveness (as compared to meat from livestock) [40].

The results of our study confirm the findings of existing studies on consumer preferences for insects and the attitudinal and socioeconomic profile of early adopters. Yet, different to the existing literature, we offer a specific insect based product, namely an insect fortified burger, instead of asking the respondents to state their general willingness to consume insects. We believe this makes it easier for the respondents to state their preferences. An additional feature of this study is the possibility to distinguish a consumer's willingness to consume insects and her interest to consume environmentally friendly products. Our results indicate that the willingness to consume insects is largely connected to the demand for products low in carbon emissions. Hence, this finding implies that there is a positive and substantial willingness-to-pay for products with a low carbon footprint, regardless of the nature of the food product as meat or meat substitute.

Last, we do not find a significant effect of the provision of prior information to the respondents. A possible explanation is that consumers are already aware that cultivating and eating insects could provide a solution to environmental and health problems. This is in line with the findings by Hoeck et al. [41]. An implication for adverting insect-based food products would be to concentrate on highlighting the negative consequences of meat consumption on climate and health instead of praising insects as a solution to the problem. However, this requires further research.

**Author Contributions:** Conceptualization, M.S. and S.V.; methodology, L.K.; formal analysis, L.K.; data curation (and data acquisition and analysis), L.K. and S.V.; writing—original draft preparation, L.K.; writing—review and editing, M.S. and S.V.; supervision, M.S.; project administration, S.V.

**Funding:** The work was developed out of the Master thesis of Ms. Saskia Vetter at the Agricultural Faculty of Kiel University and was completed at a time when all authors were employed at Kiel University.

**Acknowledgments:** We acknowledge financial support by Land Schleswig-Holstein within the funding program Open Access Publikationsfond.

**Conflicts of Interest:** The authors declare no conflict of interest.

## Appendix A

**Table A1.** Construction of attitudinal variables.

| | **Sie Sind Gefährlich Und Können Krankheiten Übertragen** | **They are Dangerous and can Transmit Diseases** |
|---|---|---|
| Aversion Insects | Insekten zu essen ist primitiv | Eating insects is atavistic |
| | Insekten sind nicht essbar | Insects are not edible |
| Disgust | Insekten sind eklig | Insects are disgusting |
| Food trend | Ich besuche gerne Street Food Festivals | I like visiting street food festivals |
| | Ich lese gerne food blogs im Internet | I like reading food blogs in the internet |
| | Ich besuche gerne Kochkurse | I like attending cooking classes. |
| Food neophobia | Ich probiere gerne neue Lebensmittel | I like testing new food products |
| | Ich kaufe und esse gerne exotische Lebensmittel | I like buying and eating exotic food products |
| | Ich kaufe und esse mir unbekannte Lebensmittel | I like buying and eating food products that are unfamiliar to me. |
| | Ich halte Ausschau nach Zubereitungsweisen für ungewöhnliche Gerichte | I watch out for new recipes and exceptional dishes. |
| | Rezepte und Artikel über Lebensmittel aus anderen kulinarischen Traditionen bringen mich dazu, in der Küche zu experimentieren | Recipes and articles about food products out of different traditions make me do experiments in the kitchen. |
| | Ich mag alles was meine Essgewohnheiten verändern könnte | I like everything that changes my eating habits. |
| Attitude Bio | Die Natürlichkeit der Produkte ist für mich ein wichtiges Qualitätskriterium | The naturalness of the products is an important quality attribute to me. |
| | Ich versuche möglichst Lebensmittel aus biologischem Anbau zu kaufen | I try to purchase organic products. |
| | Ich versuche Produkte mit Zusatzstoffen zu vermeiden | I try to avoid products with food additives. |
| | Die Zertifizierung der Qualität ist mir wichtig. | Quality certification is important to me. |
| | Die natürliche Produktionsweise ist mir wichtig | The naturalness of the production methods is important to me. |
| Convenience | Wie häufig werden die folgenden Lebensmittel konsumiert: | How often do you consume? |
| | Fertigprodukte | Convenience product |
| | Tiefkühlprodute | Frozen food |

**Table A2.** Willingness-to-pay estimates.

| Attribute | Class 1 | Class 2 | Class 3 |
|---|---|---|---|
| bio | n.s. | n.s. | +5.50 *** |
| conventional | n.s. | n.s. | +3.23 *** |
| pic_g | +4.97 ^ | n.s. | n.s. |
| pic_ug | +3.69 ^ | n.s. | n.s. |
| hc | n.s. | n.s. | +4.39 *** |
| no health claim | n.s. | n.s. | +3.80 *** |
| pat_i | −0.61 ** | n.s. | +4.60 * |
| pat_b | 8.62 ** | n.s. | +4.13 * |
| co2_3 | n.s. | n.s. | +2.87 *** |
| co2_2 | n.s. | n.s. | +4.80 *** |
| co2_1 | +5.38 * | n.s. | +5.43 *** |

Note: $* p < 0.10$, $** p < 0.05$, $*** p < 0.01$, $\hat{} \ p < 0.02$.

*Technical Appendix: LCM Model Selection*

To detect the best specifications, models with one up to five classes are estimated using Stata 13. The models were estimated with and without the membership variables and both the Bayesian Information Criterium (BIC) and the bias corrected Akaike Information Criterium (cAIC) are computed and showed in Table A3 for all four models.

Both BIC and cAIC are frequently used to determine the appropriate number of latent classes observed in the data [31]. While the log likelihood of the models increases with a larger number of classes, two classes provide the best fit using the cAIC, while according to the BIC, three classes show the best fit. Both LCM models with two and three classes are preferable to the conditional logit, which is tested by the likelihood ratio test. The respective values are 2138 for two classes and 2246 for three classes with the respective critical values of $\chi^2_{(17,0.01)} = 33.41$ and $\chi^2_{(40,0.01)} = 63.7$. As both cAIC and BIC suggest a different number of classes, we opt for the parsimonious model and estimate the LCM with two classes in this analysis.

**Table A3.** Goodness of fit for model selection.

| Number of Classes | Log Likelihood | Number of Parameters | cAIC | BIC |
|---|---|---|---|---|
| 2 | −1659.652 | 27 | 3496.443 | 3469.443 |
| **3** | **−1492.286** | **46** | **3286.363** | **3240.363** |
| 4 | −1439.01 | 65 | 3304.465 | 3239.465 |
| 5 | −1391.424 | 84 | 3333.946 | 3249.946 |

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
