# Peer review of "Disgusting or Innovative-Consumer Willingness to Pay for Insect Based Burger Patties in Germany"

_sustainability, doi:10.3390/su11071878_

Round 1
Reviewer 1 Report
Manuscript (MS): Disgusting or Innovative-Consumer Willingness to Pay (WTP) for Insect Based Burger Patties in Germany
MS submitted to the journal “Sustainability”.
The MS reflects relevant information and may be accepted after the comments are addressed by improving the paper.
1. This study analysed the preferences of German consumers for insect-based products to identify whether they are willing to adopt insects into their diet. For this purpose, an online based study was conducted in 2016, in which respondents chose between an ordinary burger and a burger with a beef burger patty fortified with insect flour. Consumers those who prefer to consume insects in their diet, perceive insects as healthy due to their nutritional value and high concentration of omega 3 fatty acids, however, the lack of food safety standards of insect feeds are associated with the risk of diseases and illnesses. Authors have estimated a latent class model to link the observed choice behaviour to socioeconomic and attitudinal characteristics. The empirical model specifically accounts for heterogeneity in choice behaviour by providing group specific WTP estimates. This is a rather new and unique approach on estimating WTP.
2. The first place “choice experiment” (CE) appear in the paper is the experimental design. It was not mentioned in the title, abstract or introduction. Participants of the CE were asked to choose between two different burger alternatives and a no choice alternative, which was shown in Figure one. Each alternative was characterized by six attributes: production method, health claim, visual impression, carbon emissions, composition of the burger patty, and the purchase price. Figure one was presented in German language, and also not sure whether all attributes were presented to the enumerators. However, in Table one all attributes were presented. In Table one, two variables were used for carbon emission attribute, but not explained the meaning of the two variables, co2_1 and co2_2. What are co2_1 and co2_2? The emission co2_1 and co2_2, may be the indication of high and low emission, which was not clearly explained in the text.
In this CE study choice has 4 attributes with 2 levels, which creates possible 24 = 2x2x2x2 = 16 possible combinations and 2 attributes with 3 levels, which creates 32=3x3=9. Hence, the total combinations are 16x9=144, which is correct. However, if you add co2_2, this creates another set of 9 which will add to the total of 144x9= 1296. The total you used as 144, is a wrong number.
Suggestion 1: Please correct the total number of possible combination used in your model depending on the attributes and levels you used in the study. Please translate figure one into English. If you describe your analysis as a CVM, these suggestions are not necessary.
3. WTP function was estimated using a conditional logit Latent Class Model (LCM) which is similar to the Random Parameters Logit (RPL) model. While the RPL assumes a continuous distribution of the parameters to introduce heterogeneity, the LCM uses discrete classes to reach the same. The LCM is a special case of the RPL with parameters being distributed discretely and hence can be referred to as a semi-parametric sister of the RPL. Hence, the results shown in table 3 are binary choice logit model (ML estimates), which is the answer to WTP of Contingent Valuation Method (CVM), which cannot be expressed as a CE results. Equation (1) is a binary choice function.
CE and CVM results are more or less the same. However, estimation procedures are quite different. If it uses in a wrong perspectives, the reader of the manuscript will get confused. The superior estimation efficiency and the greater explanation provided by the CE, indicates the greater capacity of CE to allow an understanding of the choices of respondents.
Suggestion 2: Please re-write the estimation procedure very clearly whether it is a CVM estimation of WTP or CE to derive economic values. Please do not confuse reader by mixing of all valuation methods.
4. In the discussion section you indicate that your results are CE based. This is wrong. You should combine your choice set shown in Figure one in the analysis if it is a CE based.
Suggestion 3: Please correct.
5. Conclusion and implications section is very short yet mention about CE. Please correct.
Congratulations and best wishes to authors for their attempt to bring this research to the public domain.
Author Response
The MS reflects relevant information and may be accepted after the comments are addressed by improving the paper.
1. This study analysed the preferences of German consumers for insect-based products to identify whether they are willing to adopt insects into their diet. For this purpose, an online based study was conducted in 2016, in which respondents chose between an ordinary burger and a burger with a beef burger patty fortified with insect flour. Consumers those who prefer to consume insects in their diet, perceive insects as healthy due to their nutritional value and high concentration of omega 3 fatty acids, however, the lack of food safety standards of insect feeds are associated with the risk of diseases and illnesses. Authors have estimated a latent class model to link the observed choice behaviour to socioeconomic and attitudinal characteristics. The empirical model specifically accounts for heterogeneity in choice behaviour by providing group specific WTP estimates. This is a rather new and unique approach on estimating WTP.
Response 1: We thank the reviewer for this very adequate summary of the article and his interest in the work.
2. The first place “choice experiment” (CE) appear in the paper is the experimental design. It was not mentioned in the title, abstract or introduction. Participants of the CE were asked to choose between two different burger alternatives and a no choice alternative, which was shown in Figure one. Each alternative was characterized by six attributes: production method, health claim, visual impression, carbon emissions, composition of the burger patty, and the purchase price. Figure one was presented in German language, and also not sure whether all attributes were presented to the enumerators. However, in Table one all attributes were presented. In Table one, two variables were used for carbon emission attribute, but not explained the meaning of the two variables, co2_1 and co2_2. What are co2_1 and co2_2? The emission co2_1 and co2_2, may be the indication of high and low emission, which was not clearly explained in the text.
In this CE study choice has 4 attributes with 2 levels, which creates possible 24 = 2x2x2x2 = 16 possible combinations and 2 attributes with 3 levels, which creates 32=3x3=9. Hence, the total combinations are 16x9=144, which is correct. However, if you add co2_2, this creates another set of 9 which will add to the total of 144x9= 1296. The total you used as 144, is a wrong number.
Suggestion 1: Please correct the total number of possible combination used in your model depending on the attributes and levels you used in the study. Please translate figure one into English. If you describe your analysis as a CVM, these suggestions are not necessary.
Response 2a: We thank the reviewer for the observation. Indeed, the reviewer is right. We have 4 attributes with 2 levels and 2 attributes (price, CO2) with 3 (0.71, 1.24, 2.48; 1 car, 2 cars, 3 cars) levels, which adds up to 144 possible combinations. co2_1 and co2_2 are different levels of the same attribute. We included them as individual dummies in the regression to allow for non-linear effects, but they were displayed jointly in the attribute CO2 to the respondents. Thus, 144 is the actual number of possible combinations. To avoid confusion about the variables used in the latent class model estimation we moved Table 1 (which is now Table2) to the statistical modeling section. Table 2 also shows how the inclusion of the no choice option chances the coding of the variables in the econometric estimation.
Response 2b: We very much agree and included an English translation instead of the German version in the paper.
Response 2c: We thank the reviewer for the observation. We included a sentence about the choice experiment on page 4. Indeed, the choice experiment was used to elicit the willingness-to-pay estimate, while no CVM is employed. The binary choice models estimated are used to inform the choice of the variables used in the latent class estimation.
3. WTP function was estimated using a conditional logit Latent Class Model (LCM) which is similar to the Random Parameters Logit (RPL) model. While the RPL assumes a continuous distribution of the parameters to introduce heterogeneity, the LCM uses discrete classes to reach the same. The LCM is a special case of the RPL with parameters being distributed discretely and hence can be referred to as a semi-parametric sister of the RPL. Hence, the results shown in table 3 are binary choice logit model (ML estimates), which is the answer to WTP of Contingent Valuation Method (CVM), which cannot be expressed as a CE results. Equation (1) is a binary choice function.
CE and CVM results are more or less the same. However, estimation procedures are quite different. If it uses in a wrong perspectives, the reader of the manuscript will get confused. The superior estimation efficiency and the greater explanation provided by the CE, indicates the greater capacity of CE to allow an understanding of the choices of respondents.
Suggestion 2: Please re-write the estimation procedure very clearly whether it is a CVM estimation of WTP or CE to derive economic values. Please do not confuse reader by mixing of all valuation methods.
Response 3: We thank the reviewer for pointing out the ambiguity regarding the willingness-to-pay estimation. Indeed, the WFP was computed based on the choice experiment. No CVM was used to compute WTP estimates. The binary choice model, described in equation (4), was used to inform and justify the choice of the membership variables and to compute predicted probabilities. We extended the statistical modeling section to be clearer about the econometric strategy and the econometric model used for the WTP estimation. In addition to that, we added footnote 4 to make clear that the CVM was not used for WTP estimation. We also added an explanation for equation (1).
4. In the discussion section you indicate that your results are CE based. This is wrong. You should combine your choice set shown in Figure one in the analysis if it is a CE based.
Suggestion 3: Please correct.
Response 4: As explained in response 3 the WTP estimate are based on the CE.
5. Conclusion and implications section is very short yet mention about CE. Please correct.
Response 5: We agree with the reviewer that the conclusion and implication section is short. This is partly due to the structure of the discussion and conclusions section which entail the discussion of the results and the conclusion in separate subsections. Nevertheless, we amended and included an additional paragraph on conclusions and implications.
Reviewer 2 Report
Eating insect-based food is very common in certain regions, but it is interesting to see a study conducted in a developed country. Overall, this is a well-developed study. I have a few comments and questions for the authors to consider and respond.
1. It might be more accurate to use “willingness to accept” rather than “willingness to pay” in this case, since consumers only would like to accept it with a discount.
2. I didn’t find any results surprising or beyond my expectations, and they are pretty consistent with previous findings. Therefore, the authors may want to emphasize more on their contributions and implications of this study.
3. Regarding the experimental design
a. The authors impose that the consumers couldn’t distinguish the burgers by visual inspection, how about taste? Won’t consumers suspect the taste would be different, which could influence their decisions too?
b. As the authors noted, there are a few unrealistic attribute combinations. I believe that these combinations can be restricted when design the choice experiments, but I understand the step cannot be modified now.
c. The authors provided several information treatments. However, did the authors measure respondents’ prior knowledge and experience on insect-based food before these treatments? Do these factors make any differences?
4. Regarding the data, the respondents were contacted via social media and email. Although a national reprehensive sample is not the goal, how did the authors control the quality of the sample? What are the criteria? Is there any concern on selection bias?
5. In the statistical modeling, please define all variables and parameters. For example, in eq 2), the WTP is defined as –r/b, and r is the coefficient for attribute, then what is b here?
6. Eating insect-based food could be a hypothetical scenario for many respondents in Germany. How did the authors control the hypothetical bias?
Author Response
Eating insect-based food is very common in certain regions, but it is interesting to see a study conducted in a developed country. Overall, this is a well-developed study. I have a few comments and questions for the authors to consider and respond.
1. It might be more accurate to use “willingness to accept” rather than “willingness to pay” in this case, since consumers only would like to accept it with a discount.
Response 1: We thank the reviewer for this suggestion. We generally agree with the reviewer and we included the interpretation as willingness to accept, as the negative willingness to pay, in the statistical modeling section. We also use the term willingness to accept in the discussion of the results on page 28. We prefer to keep the general term willingness to pay as it is the usual keyword in studies on consumer preferences.
2. I didn’t find any results surprising or beyond my expectations, and they are pretty consistent with previous findings. Therefore, the authors may want to emphasize more on their contributions and implications of this study.
Response 2: We thank the reviewer for sharing the concern. We agree that the study confirmed previous research, which we see as an endorsement of our study. We believe that research on consumer preferences for insect-based product is still at an early stage and that more research is needed to confirm the existing hypotheses. In addition to that, our study is the first to estimate economic willingness to pay estimates for Germany and the first to consider a fortified product (which is now available in the market) instead of asking respondents for their general willingness to consume insects. To address the concern of the reviewer, we added a paragraph on our contributions in the conclusion section.
3. Regarding the experimental design
a. The authors impose that the consumers couldn’t distinguish the burgers by visual inspection, how about taste? Won’t consumers suspect the taste would be different, which could influence their decisions too?
Response 3a: We agree with the reviewer that taste plays an important role. However, in this experiment it was not possible to include sensory aspects. This should be considered for future research and is included in the last paragraph of the limitations and future research section.
b. As the authors noted, there are a few unrealistic attribute combinations. I believe that these combinations can be restricted when design the choice experiments, but I understand the step cannot be modified now.
Response 3b: The reviewer is correct that some attribute combinations are unrealistic. However, we choose to include the unrealistic combinations in order to be able to distinguish between the respondents’ preferences for low CO2 emissions and the type of burger.
c. The authors provided several information treatments. However, did the authors measure respondents’ prior knowledge and experience on insect-based food before these treatments? Do these factors make any differences?
Response 3c: We very much agree with the reviewer that it is important. We tested for the relevance of both the information treatment and past experience with insect consumption in the latent class model. Both were not significant in determining class membership. As it is preferable to use a parsimonious latent class model, we omitted the two variable from the final regressions. As the importance was underlined by the reviewer, we added a sentence at the bottom of page 21.
4. Regarding the data, the respondents were contacted via social media and email. Although a national reprehensive sample is not the goal, how did the authors control the quality of the sample? What are the criteria? Is there any concern on selection bias?
Response 4: We thank the reviewer for pointing at the possible self selection problem. The participants knew that the surveys was about novel foodstuffs, but they were not aware about the type of product offered in the choice experiment. Due to a relatively large number of vegetarians, we believe that there is little selection bias. Assessing the quality of the sample is difficult since we aimed at so-called early adopters and it is difficult to sample according to these characteristics. Hence, we trust in the quality of the sample but name it as the major shortcoming of this study.
5. In the statistical modeling, please define all variables and parameters. For example, in eq 2), the WTP is defined as –r/b, and r is the coefficient for attribute, then what is b here?
Response 5: We thank the reviewer for this observation. Indeed, the description of b was missing. The section was amended to avoid confusion between the models, as requested by another reviewer. The explanation of b is now available in the paragraph before equation (4) which is the previous equation (2).
6. Eating insect-based food could be a hypothetical scenario for many respondents in Germany. How did the authors control the hypothetical bias?
Response 6: We agree with the reviewer that the choice situation is hypothetical. At the time of the experiment, the insect burger was not available in German supermarkets. To reduce the hypothetical bias, we included a standard statement encouraging the respondents to act as in a real situation without having the fear to answer a question incorrectly.
Reviewer 3 Report
I've read the paper with interest and I support its publication in Sustainability. The methodology is rigourous and results are clearly presented. The choice of using a latent class model is particularly sound.
The limitation stands in the small sample analyzed, as it is recognized also by authors themselves. However I think the paper is valuable as it stimulates discussion on this issue and provides an interesting methodological basis for future research aiming to extend and improve the sample selection.
Author Response
I've read the paper with interest and I support its publication in Sustainability. The methodology is rigourous and results are clearly presented. The choice of using a latent class model is particularly sound.
The limitation stands in the small sample analyzed, as it is recognized also by authors themselves. However I think the paper is valuable as it stimulates discussion on this issue and provides an interesting methodological basis for future research aiming to extend and improve the sample selection.
Response 1: We thank the reviewer for her/his kind words and took the suggestion to include future research possibilities in the discussion and conclusion section.
Round 2
Reviewer 2 Report
The authors have provided sound explanations for my concerns, and I don't have further questions.